# Formulation and Evaluation of Novel Additive-Free Spray-Dried Triamcinolone Acetonide Microspheres for Pulmonary Delivery: A Pharmacokinetic Study

**DOI:** 10.3390/pharmaceutics14112354

**Published:** 2022-10-31

**Authors:** Mohammed Amran, El-Sayed Khafagy, Hatem I. Mokhtar, Sawsan A. Zaitone, Yasser M. Moustafa, Shadeed Gad

**Affiliations:** 1Department of Pharmacy, Faculty of Medicine and Health Sciences, Thamar University, Thamar 425897, Yemen; 2Department of Pharmacy, Al-Manara College for Medical Sciences, Maysan 62001, Iraq; 3Department of Pharmaceutics, College of Pharmacy, Prince Sattam Bin Abdulaziz University, Al-Kharj 11942, Saudi Arabia; 4Department of Pharmaceutical Chemistry, Faculty of Pharmacy, Sinai University, Kantra 41636, Egypt; 5Department of Pharmacology and Toxicology, Faculty of Pharmacy, University of Tabuk, Tabuk 71491, Saudi Arabia; 6Department of Pharmacology and Toxicology, Faculty of Pharmacy, Suez Canal University, Ismailia 41522, Egypt; 7Department of Pharmacology and Toxicology, Faculty of Pharmacy, Badr University in Cairo, Cairo 11829, Egypt; 8Department of Pharmaceutics and Industrial Pharmacy, Faculty of Pharmacy, Suez Canal University, Ismailia 41522, Egypt

**Keywords:** blowing agent, microspheres, pulmonary delivery, rat, spray drying, triamcinolone acetonide

## Abstract

This work aimed to establish a simple method to produce additive-free triamcinolone acetonide (TAA) microspheres suitable for pulmonary delivery, and therefore more simple manufacturing steps will be warranted. The spray-drying process involved the optimization of the TAA feed ratio in a concentration range of 1–3% *w*/*v* from different ethanol/water compositions with/without adding ammonium bicarbonate as a blowing agent. Characterization of the formulas was performed via scanning electron microscopy, Fourier-transform infrared spectroscopy, differential scanning calorimetry, and powder X-ray diffraction. Our results indicated that the size and morphology of spray-dried TAA particles were dependent on the feed and solvent concentrations in the spray-dried formulations. Furthermore, adding the blowing agent, ammonium bicarbonate, did not produce a significant enhancement in particle characteristics. We prepared additive-free TAA microspheres and found that TAA formulation #1 had optimal physical properties in terms of diameter (2.24 ± 0.27 µm), bulk density (0.95 ± 0.05), tapped density (1.18 ± 0.07), and flowability for deposition during the pulmonary tract, from a centric airway to the alveoli as indicated by Carr’s index = 19 ± 0.01. Hence, formulation #1 was selected to be tested for pharmacokinetic characters. Rats received pulmonary doses of TAA formula #1 and then the TTA concentration in plasma, fluid broncho-alveolar lavage, and lung tissues was determined by HPLC. The TAA concentration at 15 min was 0.55 ± 0.02 µg/mL in plasma, 16.74 ± 2 µg/mL in bronchoalveolar lavage, and 8.96 ± 0.65 µg/mL in lung homogenates, while at the 24 h time point, the TAA concentration was 0.03 ± 0.02 µg/mL in plasma, 1.48 ± 0.27 µg/mL in bronchoalveolar lavage, and 3.79 ± 0.33 µg/mL in lung homogenates. We found that TAA remained in curative concentrations in the rat lung tissues for at least 24 h after pulmonary administration. Therefore, we can conclude that additive-free spray-dried TAA microspheres were promising for treating lung diseases. The current novel preparation technology has applications in the design of preparations for TAA or other therapeutic agents designed for pulmonary delivery.

## 1. Introduction

Respiratory illnesses are usually treated by direct inhalation of drug formulations [1,2]. The inhalation route is considered superior to the oral route because of its rapid onset, bypassing the first-pass metabolism, and decreased incidence of adverse effects [3,4]. In recent years, pulmonary drug delivery has shown promising results in systemic drug delivery [5]. The human lung is an effective gate for drug entry to circulation; this is attributed to the lung’s huge available surface area for absorption (hundred-meter square), very tinny absorption surface, and high blood flow (5 L per min). All these factors result in rapid drug onset and distribution without loss through first-pass hepatic metabolism, thus leading to high bioavailability [6]. 

Microspheres are spherical microparticles with a diameter ranging between 100 nm and 100 µm. Due to their dimensionality, microspheres of drug formulations are suitable candidates for administration via the inhalation route. Several methods were applied for microspheres preparation, such as a non-solvent addition (coacervation), solvent partitioning, the solvent evaporation technique, the supercritical fluid extraction method, and spray-drying. 

Spray-drying is a potential method for manufacturing microspherical particles. Previous publications described novel spray-drying methods to prepare nanoporous microspheres [7,8]. Some of the reported methods designed the produced particles to be free from additives [9]. In general, the described methods involved the utilization of co-solvents and blowing agents (e.g., ammonium carbonate) in producing nanoporous microsphere dry powders suitable for drug delivery. This provided benefits of spray-drying over ordinary methods as it involved an easy one-step particle engineering process in producing microspheres suitable for inhalation. The volatile property of the blowing agent suggests that they might be removed during spray-drying. Upon testing with a twin-stage impinger as an in vitro model for the human pulmonary system, the microspheres prepared by this method resided in the impinger lower region, indicating a respirable formulation. In addition, the fabricated powders had a good flow property and, consequently, had the potential for application in the production of dry powder inhalers [10].

Triamcinolone acetonide (TAA) is a glucocorticoid used as an anti-inflammatory and immunosuppressive agent [11]. TAA is a hydrophobic compound with low solubility in water (14–20 ug /mL) [12]. According to the literature survey and until the start of this work, no previous work had been published applying and evaluating spray-drying for preparing TAA nonporous microspheres as a potential inhaler formulation. Different formulations were engineered using different TAA feed concentrations (1–3%), and an ethanol–water mixture was used as a solvent in different ratios [13]; this led to the production of TAA particles with a wide range of morphological characteristics.

In this study, we examined the properties of spray-dried TAA nonporous microsphere particles formulated with/without adding a blowing agent, ammonium bicarbonate, and investigated the impact of the process parameters on the physicochemical properties of the prepared TAA microsphere formulas. The formula with the best physiochemical characteristics was selected to be tested in vivo for its pharmacokinetic behavior and, hence, to determine its utility to be applied in pulmonary drug delivery.

## 2. Materials and Methods

### 2.1. Materials

Micronized TAA was purchased from Newchem S.p.A. (Milano, Italy, Bach No.: NT2350 M). Ammonium bicarbonate (ABC) powder was purchased from El-Nasr Pharmaceutical Chemical Co. (Cairo, Egypt). Ethanol 99.9% was provided by Lab-Scan (Gliwice, Poland). Liquid nitrogen was purchased from Helwan for industrial gases (15 of May City, Egypt). Deionized water was supplied by a Purite Prestige Analyst HP water distillation apparatus.

### 2.2. Methods

#### 2.2.1. Spray Drying of Triamcinolone Acetonide

Different formulations with TAA feed concentrations equal to 1–3% were dissolved in ethanol–water (solvent) in different ratios (70–90% *v*/*v*) [13]. These particular spray-drying conditions (feed concentration and solvent concentration) were chosen to yield TAA particles with a wide range of morphological characteristics.

The prepared mixtures were spray dried by the Büchi B-290 Mini Spray Dryer, Büchi (Flawil, Switzerland) operated in the open mode configuration [14] using air as the drying medium. The resulting spray-dried powders were collected using a small synthetic bristle brush from the instrument’s collection vessel. Collected samples were stored in screw-capped glass jars or clear glass sample tubes in a sealed desiccator at 4 °C with drying silica gel [15]. The production yield of each prepared TAA formulation was calculated (as weight%) by dividing the collected powder by the initial feed quantity [16].

#### 2.2.2. Differential Scanning Calorimetric Analysis (DSC) 

Samples (3–7 mg) were accurately weighed using ENTRIS224-1S Balance, Sartorius Lab Instrument GmbH & Co. K.G. (Gottingen, Germany), into sealed aluminum pans with a capacity of 40 μL. The weighing pans were hermetically sealed with three vent holes. The samples were scanned at a heating rate of 10 °C/min under nitrogen purge from 25–400 °C [17] using Shimadzu DSC-60 (Kyoto, Japan) calorimeter. Shimadzu DSC-60 data analysis software was used for the analysis of thermodynamic events.

#### 2.2.3. Fourier Transform Infrared Spectroscopy (FTIR)

Infrared absorption spectra of the prepared samples were scanned over a wavenumber range of 4000–400 cm^−1^ [18] by 4100 JASCO FTIR-4100, JASCO International Co., Ltd. (Tokyo, Japan) at the Micro-Analytical Center of the Faculty of Science (Cairo University, Egypt). Samples were scanned as KBr discs prepared with 1 mg % sample loading. Samples to be examined were ground with potassium, and then the mixture was placed in KBr die and compressed by the application of 8 tons of pressure in an I.R. press. 

#### 2.2.4. Scanning Electron Microscopy (SEM)

Visualization of the spray-dried sample morphology was performed by scanning electron microscopy (SEM) [16] using the Quanta FEG 250 (FEI Co., Eindhoven, Eindhoven, The Netherlands) variable pressure microscope at the Egyptian Mineral Resources Authority (Giza, Egypt). The samples were gold-sputtered while fixed on an aluminum stub with a gold film thickness of ten nm using the EMITECH K550X sputter coater, Quorum Technologies Ltd. (Lewes, UK) before microscopy. The sample images were obtained via the acquisition of secondary electrons and visualized at different magnifications. Particle size distribution and the average particle diameter for each sample were calculated from the acquired SEM images [19]. 

#### 2.2.5. Powder X-ray Diffraction 

The X-ray Diffraction (XRD) value of the TAA powder was performed using samples that contained cavities 0.5-mm in depth and 9-mm in diameter in a low background silicon holder (Empyrean, Malvern PANalytical, Malvern, UK). A Siemens D500 powder Diffractometer, Siemens (München, Germany), equipped with a DACO MP wide-range goniometer with dispersal, anti-scatter, and receiving slits at 1.0°, 1.0°, and 0.15°, respectively, was applied. Monochromatic Cu Kα X-rays were produced by a Cu anode X-ray tube at 40 kV and 30 mA with a Ni filter. Qualitative analysis measurements were performed in the ranges of 5° to 40° or 5° to 80° on the 2θ scale with a step size of 0.05°/s [20]. 

#### 2.2.6. Density and Compressibility Measurements

Bulk density (ρb) was calculated by weighing the powder sample required to fill 1ml in a 10-mL graduated cylinder and calculating the powder weight ratio to the measured bulk volume of the sample [21]. The tapped density (ρt) of processed powder was then determined by tapping the cylinder containing the powder 100 times onto a horizontal surface from a 1 inch height as described previously [9]. The applied height for performing tapped density was according to previous works on spray-dried products [9] and was fixed for all the tested formulations to enable their comparison. 

The tapped density, ρt, was then calculated as the mass ratio to the sample tapped volume [21]. 

The compressibility of powdered samples was determined as Carr’s index from tapped and bulk through the following equation [22]:$$\mathrm{Carr}’s\;\mathrm{index}\;(\%)=\frac{\left(\rho_{t}-\rho_{b}\right)}{\rho_{t}}\times 100\%$$

#### 2.2.7. Chromatographic Analysis of TAA

HPLC chromatographic separation of TAA powder samples was performed using the kromasil^®^ C18 5 µ, 150 × 4.6 mm column (Nouryon, Göteborg, Sweden) as a stationary phase. The chromatographic analysis was performed using the YL9101 HPLC system (YOUNGIN Chromass corporation, Gyeonggi-do, South Korea) composed of a YL9110 quaternary pump, a YL9101 inline vacuum degasser, a YL9131 thermostated column compartment, a YL9120 UV/Vis. variable wavelength detector, and a 7725i Rheodyne^®^ manual injector. Autochrom3000 software was used for instrument control and chromatographic data acquisition and processing. The mobile phase consisted of acetonitrile:water (45:55 *v*/*v*) pumped at a flow rate of 1 mL/min. The detection wavelength was fixed at 252 nm (λmax for TAA) [23]. The injection volume was 20 µL.

### 2.3. In Vivo Direct Pulmonary Delivery of TAA Microspheres 

These experiments follow the guidelines for the use and care of laboratory animals and were approved by the research ethics committee in the Faculty of Pharmacy, Suez Canal University (approval # 3102015MA01). Male albino rats were purchased from the National Organization for Biological Products and Vaccines (Cairo, Egypt) and housed at the animal house of the Faculty of Pharmacy, Suez Canal University. Rats were healthy, showed no clinical indications of sickness, and weighed 200–250 g at the beginning of the experiment. The temperature in the animal room was adjusted to 25 ± 5 °C in a normal light/dim cycle. Rats were permitted to access water and food ad libitum throughout the course of the experiment, and every effort was performed to minimize animal suffering. Rats were randomly allocated to 13 groups (*n* = 3) and anesthetized by intraperitoneal injection of 80 mg/kg of ketamine (Pharco Pharmaceuticals, Alexandria, Egypt) and xylazine (10 mg/kg) (Nembutal-Dainippon Sumitomo Pharma, Osaka, Japan) [24]. This dose was selected from a previous study [25]. A dose of 80 mg/kg of ketamine and 10 mg/kg of xylazine was not reported to affect body organs (liver, kidney, heart, and lungs) [25].

Before starting the TAA pulmonary delivery procedures, the formulations were numbered, and 5 mg of microsphere particles of TAA powders were weighed out into hydroxypropyl methylcellulose capsules and put away at 4 °C preceding dosing. After anesthesia, rats were set onto an inclination board to reorganize the visual position of the insertion cannula into the trachea utilizing an 18-gage needle connector for the insufflation of medication [26] followed by consequent infusion and retrieval of broncho-alveolar lavage (BAL) fluid. Furthermore, the tip of the insufflation cannula was put in the trachea just over the carina, and 5 mg [27] of TAA microsphere powder was conveyed across the insufflation cannula by quickly pushing a 3-cm air bolus across the gadget 3 times. Then, the pharmacokinetic study was immediately launched [28]. 

#### 2.3.1. Blood Sample Collection 

Blood samples were withdrawn through the rat retro-orbital plexus and collected at 0 (before dose) and, following respiratory insufflation, at 0.25, 0.5, 1, 2, 3, 4, 8, 10, 12, 16, 20, and 24 h. Approximately 0.4 mL of blood was collected at every time point, and samples were collected in centrifuge tubes containing EDTA. Then, plasma samples were isolated after centrifugation (Xianhe, Model: 800, Changsha, Hunan, China) at 6000 rpm for 30 min. After that, the obtained plasma samples were frozen at −80 °C [29].

#### 2.3.2. Fluid Broncho-Alveolar Lavage Sample Collection 

For analysis of the level of TAA in the lung lining, rats were euthanized by cutting the spinal cord, exsanguinated via cardiovascular cut, and BAL was performed on the residue of lung lavage buffer [30] composed of 8.5 g of NaCl, 0.37 g of KCl, 0.26 g of NaH2PO4, and 1 g of glucose in 1 L of distilled water. After surgery, 1 mL of the lavage buffer was gradually embedded through a 5-mL syringe, and the maximum amount of the liquid was collected for the ensuing examination. The recovered BAL liquid was centrifuged at 1060 rpm for 2 min [27], and then the supernatant was kept at −80 °C [31].

#### 2.3.3. Lung Tissue Sample Collection 

After BAL, the lungs were isolated from the rat chests, weighed, and put into a 15 mL tube; then, they were snap-frozen on dry ice and kept at −80 °C until homogenization. For homogenization, lung tissues were mixed with 3.0 mL of the lavage buffer in a 15-mL tube; then lungs were homogenized utilizing a Teflon homogenizer and centrifuged at 14,000 rpm for 2 min [27]. 

#### 2.3.4. Analysis of TTA Concentration in the Biological Samples

Aliquots of 200 μL of the collected plasma, 500 μL of BAL samples, or lung tissue slurry samples were mixed in a small glass tube with 3 mL of a mixture of ethyl acetate: methylene chloride: tertiary-butyl ether (4:3:3; *v*/*v*/*v*) [32]. The mixture was mixed by vortex for 5 min and then centrifuged for 15 min at 14,000 rpm. The organic layer for each test was transported to a clean test tube and evaporated to dryness at 50 °C under a nitrogen atmosphere. The residual deposits after sample drying were re-dissolved into 500 μL of the applied HPLC mobile phase, filtered with a 0.45 µm PTFE membrane syringe filter, and then injected into the applied HPLC system (described in Section 2.2.7).

## 3. Results and Discussion

### 3.1. Spray-Drying of TAA Formulations from Ethanol without Blowing Agent 

In the context of the aim of the current work, the feasibility of the production of a respirable TAA spray-dried product was investigated. The spray-dried formulations had different concentrations of TAA in ethanol:water mixtures of different ratios as detailed in Table 1. The prepared spray-dried products were then characterized by performing SEM, XRD, FTIR, and DSC, as well as physical measurements of bulk and tapped densities and calculation of Carr’s index on the products.

The aim of these steps was to investigate the most appropriate formula and process parameters that enable the production of a respirable spray-dried product. The controllable formulation input parameters were the TAA feed concentration and the solvent composition expressed by the percentage of ethanol. The impact of the described input parameters on the properties of the spray-drying powders was investigated by the described characterization methods. Briefly, SEM enabled the evaluation of the size and morphology of the resulting products to enable the selection of spray-dried powders of the lowest particle size and highest porosity. DSC and XRD indicate the crystallinity of the products while FTIR indicates the preservation of the chemical composition of the spray-dried products. Physical measurements and Carr’s index calculation enabled us to evaluate powder flowability and respirability and hence, the suitability for application in pulmonary delivery products.

The particular spray-drying conditions described in Table 1 were chosen to cover a wide range of morphological characteristics of particles created by spray-drying of TAA under different spray-drying conditions (feed concentration and solvent concentration). 

#### 3.1.1. Electron Microscopy

Electron microscopy was crucial for particle size measurement and morphology elucidation of the produced TAA spray-drying products in comparison with the unprocessed TAA raw material. The desired product was preferred to be of minimal particle size and higher surface porosity to enable higher absorption through lung tissues as well as respirability through pulmonary delivery devices. The impact of the TAA feed concentration and solvent composition on particle morphology and size was investigated as follows.

Feed concentrations were found to have a significant impact on particle morphology and size. The solid contents of TAA spray-dried formulas were examined at TAA concentrations of 1%, 1.5%, 2%, 2.5%, and 3% in the applied solvent systems. Spray-drying of the 1% (*w*/*v*) TAA formulation with 90% (*v*/*v*) ethanol (Experiment F1) resulted mainly in microsphere particles with a rough, porous surface (Figure 1B) and a diameter range of 1–5 um (Table 2) as elucidated by SEM. The observed microsphere surface roughness can improve the aerosolization capability of the spray-dried particles [33]. The obtained microspherical particles differed from the micronized crystals of unprocessed TAA applied to raw material, as shown in Figure 1A. Increasing the feed concentration of TAA from 1% up to 2.5% *w*/*v* (experiments F2–F4, Figure 1C–E) slightly increased the produced microspheres while decreasing the microsphere porosity (Table 2). 

On the other hand, spray-drying of the 3% (*w*/*v*) TAA solution from 90% (*v*/*v*) ethanol (Formulation F5) resulted in large crystalline aggregates comprised of non-porous spherical particles with a diameter size range of 1–5 µm (Table 2 and Figure 1F). It was assumed that the swelling of microspheres increased by increasing the TAA concentration, leading to an increment in the TAA network within the microspheres. These results were in accordance with Walton and Mumford’s findings [34] as they reported that increasing feed solids might lead to heavy drying chamber deposits, indicating that the maximum feed solids may have been exceeded. In addition, many spray dryers are usually operated at low feed solids and comparatively low drying temperatures as these conditions are affected by the feed properties. 

According to these results, the morphology of spray-dried TAA without the blowing agent was significantly altered depending on the feed concentration. Respirable spray-dried TAA microspheres were optimally obtained at lower feed concentrations of TAA.

The solvent composition regarding the water:ethanol ratio played an important role in determining the morphology of the produced particles. The effect of variations in the solvent composition of 70, 80, and 90% *v*/*v* ethanol was studied by formulations F6, F7, and F1, respectively (Table 1), at a 1% *w*/*v* TAA feed concentration. It was found that TAA microspheres were formed from solvent compositions of not less than 80% *v*/*v* ethanol. No microspheres were produced when the solvent composition was reduced to 70% *v*/*v* ethanol (Figure 1E,F). In addition, at co-solvent ratios of 90% *v*/*v* ethanol, the number of produced TAA microsphere particles was significantly higher than in other spray-dried formulations. Spray-drying from 80% *v*/*v* ethanol produced microsphere particles resembling those obtained from an ethanol concentration of 90% *v*/*v* ethanol, but with numerous spherical particles (Figure 1H and Table 2). Spray-drying from a solvent composition equal to 70% *v*/*v* ethanol produced irregular collapsed/doughnut-shaped porous particles (Figure 1G) with no significant change in the mean particle size compared with the product from 90% *v*/*v* ethanol (Table 2). 

According to these results, the composition of the solvent system had a significant impact on the morphology of the produced spray-dried particles as examined by SEM. Ethanol appeared to be a promising solvent for creating TAA microsphere particles by spray-drying. This was consistent with the results reported by Nolan et al., who successfully produced microsphere particles of budesonide via spray-drying from ethanolic solutions [8]. The diffusion coefficient of the solute was found to change with its concentration and composition of the solvent and thus significantly altered the resulting morphology of the spray-dried particles [35]. The process solvent was volatilized during the spray-drying process as well as any added pore-forming agent. These expanding gases were removed with the drying medium to create porous particles [36]. 

The outlet temperature was recorded for information in Table 2. The outlet temperature is defined as the air temperature of the solid particles at the cyclone entrance to the spray dryer. This temperature results mainly from the heat and mass balance within the drying cylinder, and is an aspect that cannot be directly controlled. It was reported by Prinn et al. that even at a fixed inlet temperature, the outlet temperature might vary due to differences in the solvent composition, atomizing gas flow rate, solid concentration, and liquid feed rate [37]. In addition, Euser et al. reported that air moisture might influence the outlet temperature [38]. Outlet temperature values were recorded as output parameters for TAA formulation experiments as described in Table 2. 

#### 3.1.2. Differential Scanning Calorimetry

Differential scanning calorimetry enabled the evaluation of the crystallinity of the produced spray-dried products in comparison with the starting materials and the investigation of any potential incompatibilities that may occur to the drug during the preparation process. The results of DSC indicated that neither the TAA feed concentration nor the solvent composition impacted TAA crystallinity as compared with the initial raw material, with no evidence of interactions or incompatibilities during the process. The detailed results were as follows.

Thermograms of TAA show an exothermic peak at 284.75 °C and an endothermic peak of melting at 290.25 °C, as reported by Ghanshyam et al. [17]. TAA is a thermolabile compound that melts with decomposition. Other literature values of the TAA melting point such as 284.75–290.25 °C have been reported [39,40]. The melting endotherm obtained upon examination of the unprocessed TAA starting material by DSC was in accordance with the reported data (Figure 2A). 

A Thermogram of the spray-drying product of 1% TAA from 90% *v*/*v* ethanol confirmed the crystalline nature of the TAA starting substance, as the endothermic peak observed at approximately 289 °C (Figure 2B) was characteristic of TAA melting, as confirmed by the literature values (284.75–290.25 °C) with a measured enthalpy of fusion of −73.43 J/g [17,40]. The spray-dried samples from 1% TAA in 90% *v*/*v* ethanol also showed an additional endothermic peak at 282.92 °C, attributed to melting (Figure 2B). In the DSC thermograms of the 2, 2.5, and 3% TAA spray-dried formulations, additional melting endothermic events were detected at approximately 285 °C (Figure 2C,D,F) in the product of each experiment. On the other hand, the DSC thermogram of the 1% TAA formulation spray-dried from 70% *v*/*v* ethanol showed a melting endothermic event peak at approximately 283 °C with its onset at approximately 279 °C (Figure 2G). 

In addition, a DSC thermogram of the 1% TAA formulation spray-dried from ethanol 80% *v*/*v* showed an endothermic event, peaking at approximately 286 °C with the onset at approximately 281 °C. Only one endothermic peak was recorded (Figure 2H); the sample was deemed crystalline.

Thermal analysis results suggested that spray-dried samples may be crystalline, as only one endothermic event was observed on examination of DSC scans of spray-dried samples, as summarized in Table 1. The onset of the endothermic peak of spray-dried samples was in agreement with the observed melting point of the starting material and the literature values [39,40]. The DSC of TAA spray-dried samples agreed with the results of the TAA thermal analysis [41]. This could indicate that there were no significant alterations in the thermal performance of the spray-dried samples in contrast to the starting raw material under the described conditions (Table 1). 

When DSC was performed for a spray-dried sample with heating in flowing nitrogen at 10 °C/min from 25 to 100 °C, the DSC thermogram showed weight loss in two stages. This observed weight loss for dried samples was compared to the weight loss for the starting material. DSC registered weight loss over a temperature range of 25 to 100 °C for 1, 1.5, 2, 2.5, and 3% TAA formulations spray-dried from 90% ethanol in agreement with the reported data [42]. The difference in weight loss between the starting material and spray-dried experimental samples may be due to the residual solvent content within spray-dried samples (Figure 2G). This could be justified as the boiling point of the applied ethanol–water solvent mixtures was within this temperature range.

Furthermore, the difference in weight loss between spray-dried samples may be due to the different solvent compositions applied. The higher the water content of the formulation, the more moisture is adsorbed to the product. In addition, the differences in residual solvent content between different spray-dried samples may be because of differences in the solid content of spray-dried formulations, so it is not surprising that formulations containing 2% and 3% solids retained more solvent in comparison to spray-dried formulations containing 1% total solids.

#### 3.1.3. Infrared Spectroscopy

Fourier Transform Infrared Spectroscopy (FTIR) was applied to confirm that the drug material TAA chemical structure was not impacted by the spray-drying process. From the following results, changes to the TAA feed concentration or solvent composition had no impact on the TAA chemical structure and functional groups. 

In the solid phase, the FTIR spectrum of TAA has stretching bands at 3650–3200 cm^−1^ due to –OH groups, an absorption band at 1775–1650 cm^−1^ due to the C=O group, a band at 3000–2900 cm^−1^ due to C-H bond stretching, a band at 1690–1635 cm^−1^ due to the C=C group, a band at 1310–1000 cm^−1^ due to the C-O-C group, and a band at 1050 cm^−1^ due to C–F bond stretching [39,40]. The obtained FTIR spectrum agreed with the previously reported spectrum of TAA [43]. 

The spectra of spray-dried TAA samples were identical to that of the initial material as the characteristic signals of TAA were found at the same wave numbers in the spectra of both the unprocessed raw material and the spray-dried powder samples (Figure 3A–H). This indicated that the spray-drying process did not alter the chemical structure of a compound. The lack of new bands for TAA spray-dried formulations indicated that no chemical change occurred in the active ingredient during the spray-drying process. 

#### 3.1.4. X-ray Diffraction

Confirmation of the crystalline structure and any potential polymorphs was elucidated by the application of X-ray Diffraction (XRD). XRD of TAA spray-dried formulations in comparison with the starting material indicated that the crystalline structure of the material was retained during the spray-drying process. 

The starting material of TAA was supplied as a crystalline powder, as shown by the presence of well-defined peaks in the X-ray powder graph of the TAA raw material (Figure 4A) representing a typical crystalline material. For comparison of the unprocessed TAA raw material with the spray-dried samples; it is important to note that the particle size of the TAA starting material was very large compared to spray-dried formulations. Accordingly, the starting TAA raw material was micronized (99% ≤ 5 µm) to enable comparison with the spray-dried formulations.

The peak positions and their relative intensities were consistent with the XRD spectrum (two sharp peaks at 2θ = 9.86° and 14.45°) presented by the previous literature [43]. TAA showed small intrinsic crystal peaks at 2θ of 9, 14, 17, and 24 degrees, which indicate the drug’s crystalline structure. The five sharp peaks acquired positioned for the TAA starting material at approximately 2θ of 9.8, 14.1,17, 23.8, 25.2, and 26.4 degrees were in good agreement with the previous literature for TAA raw material [43,44]. 

Analysis via XRD revealed that spray-drying of TAA from an ethanol solution resulted in crystalline powder production (Figure 4A–H). This was indicated by the XRD pattern of spray-dried samples that were typical in peak positions with the XRD graphs of the unprocessed TAA material. These results augmented the DSC findings that the spray-dried products were deemed crystalline in nature.

It is important to mention that the percentage crystallinity of spray-dried samples should be respected as relative values as the crystallinity of unprocessed TAA was assumed to be 100%. It must also be noted that amorphous phase detection by XRD is limited to 5–10% [45]. The degree of crystallinity of the spray-dried TAA formulations determined by XRD was reduced compared to raw material XRD analysis, assuming the crystallinity of unprocessed TAA was 100%. The combined area under peaks of TAA spray-dried samples was found to be reduced in comparison with that of the initial material’s XRD (Figure 4A–H). 

#### 3.1.5. Powder Physical Examination

Powder physical examination with the calculation of Carr’s index enabled the evaluation of particle flowability and hence the applicability for use in pulmonary drug delivery applications. Spray-drying was found to enhance the flowability and respirability of the TAA spray-dried products without a blowing agent. The resulting Carr’s index values were significantly impacted by the TAA feed concentration and solvent composition as highlighted in the following description of the results.

Spray-dried samples were characterized in terms of particle size, bulk density (ρb), and tapped density (ρt). All of the spray-dried formulations were compared to each other and the micronized TAA. It must be noted that the particle size of the TAA starting material was very large as compared to spray-dried formulations, so it was micronized (99% ≤ 5 µm) to enable a logical comparison with spray-dried formulations.

Particle size was determined by measuring the average diameter of 20 particles from SEM for the different TAA formulations (Table 2) and the micronized TAA. The smallest particle size was found for 1% TAA sprayed from 90% (*v*/*v*) ethanol (2.241 ± 0.35 µm), with a slight increase in the mean particle size by increasing the TAA feed concentration from 1% to 3%. For the 1% TAA spray-dried formulations sprayed from 70% ethanol, the particle size was 2.54 ± 0.32 µm, while for the 1% TAA spray-dried formulation sprayed from 80% ethanol, it was 3.12 ± 0.49 µm. Labiris and Dolovich reported that particles ranging from 1–5 μm were able to deposit in the small airways and alveoli [46]. According to the obtained results, the tested TAA formulations have particles within an acceptable particle size range for inhalation therapy.

Powders of lower density and higher flowability were observed to have better properties for application in aerosol systems [47]. The bulk and tapped densities measured for TAA formulations were compared with each other and with those measured for the micronized TAA (Table 2). Carr’s compressibility index values were also calculated for the different spray-dried TAA formulations (Table 2) to indicate powder flow properties. Carr’s index values greater than 40% are characteristic of powders of extremely poor flowability, values in the range of 32–38% are characteristic of very poor flowability, while values in the range of 25–32% are characteristic of poor flowability. Powders with good flowability are characterized by Carr’s index values below 25%. 

The calculated Carr’s index for the micronized TAA starting material was 46%, indicative of a very poorly flowing powder. The Carr’s index values calculated for spray-dried TAA formulations (Table 2) were significantly lower than that of the micronized TAA powder. The minimum values of Carr’s index were obtained from lower TAA feed concentrations (1% and 1.5%). Decreasing %ethanol in solvent composition resulted in higher Carr’s index values with increased risk of production of powders with poor flowability. These results indicate the improvement of TAA powder flowability by the spray-drying of minimal TAA feed concentration (1–1.5%) from 90% ethanol.

### 3.2. Spray-Drying of TAA Formulations with a Blowing Agent (Ammonium Bicarbonate) from Ethanol

To examine the feasibility to produce a TAA spray-dried product without the addition of a blowing agent, another set of experiments was performed with the addition of ammonium bicarbonate (ABC) as a blowing agent and a comparison of the results between the two experimental sets. Solutions of 1% and 1.5% *w*/*v* TAA with ammonium bicarbonate as a blowing agent at 10, 15, and 20% *w*/*v* were prepared in 90% (*v*/*v*) ethanol (Table 3). As with each spray-drying experiment, resultant powders were examined by SEM, DSC, FTIR, and XRD, as well as the calculation of bulk density, tapped density, and Carr’s index.

The porous particles could be produced by spray-dried products where ABC acted as the process enhancer. ABC decomposes into ammonia, carbon dioxide, and water at temperatures over 59 °C, and the ABC was removed from the spray-dried products in the exhaust gases, producing essentially pure drug particles [9].

For the formulation of the 1% solution of TAA with 10% *w*/*v* ABC spray-dried from 90% (*v*/*v*) ethanol, the SEM image (Figure 5A) consisted of rough, porous microsphere particles. Examination of the other spray-dried material by SEM revealed similar rough, porous spherical particles of TAA with the same size range (Figure 5B–D). The ability to create microsphere particles was independent of the ABC concentration employed as each formulation was seen to be morphologically similar (i.e., rough microspheres) with a similar size distribution range (Table 4). 

The impact of TAA feed concentration was also investigated; a 1.5% (*w*/*v*) TAA/10% ABC solution was also spray-dried from 90% (*v*/*v*) ethanol (Table 3). The SEM image of particles produced from this formulation showed porous, rough microspheres (Figure 5D) with a size range of 1–5 um (Table 4), indistinguishable from that obtained from the 1% TAA experiment from 10% ABC. The feed TAA concentration had no significant impact on the production of microspheres within the tested range of 1–1.5% *w*/*v*, from 10% ABC solutions in 90% ethanol. According to these results, the ABC amount and total solid concentration employed had no significant influence on the microspheres’ morphological characteristics.

Scans of TAA formulations, F8–F11, by DSC (Figure 6A–D) revealed the crystalline nature of spray-dried formulations as the thermograms displayed only one endothermic event. The obtained thermograms showed no changes in peak positions compared to the original TAA raw material. The peak obtained from spray-dried formulations with the addition of ABC became asymmetric with an irregular shape, and there were changes in the onset temperature. The endothermic melting peak was 282.92 °C for the starting material of TAA, while it was found at 84.69, 285.10, and 273.06 °C for spray-dried 1%TAA with 10%, 15%, and 20% of ammonium bicarbonate in 90% ethanol *v*/*v*, respectively. A broad endothermic peak was accompanied by weight loss, as seen in DSC scans (Figure 6A–D). It could be concluded that there were no significant differences in the thermal behavior of processed samples compared to that of initial material upon spray-drying under the described conditions. DSC scans of 1.5%TAA/10% ABC spray-dried from 90% *v*/*v* ethanol showed the same crystalline nature (Figure 6E). 

Infrared scans were evidence of the similarity between 1%TAA/ABC spray-dried formulations and the initial material as displayed (Figure 7A–D). FTIR scans were more evidence of the similarity between 1.5% TAA and 10% ABC spray-dried from 90% *v*/*v* ethanol and the initial material, as shown in Figure 7D.

The XRD scans for 1%TAA with 10%, 15%, and 20% of ABC in 90% ethanol *v*/*v* displayed a typical pattern of the crystalline material of TAA, as confirmed by the numerous diffraction peaks on the X-ray diffractogram. XRD patterns of 1% TAA/ ABC spray-dried from 90% ethanol and unprocessed material showed no difference in peak positions between the samples. However, the intensities of the peaks of TAA/ABC spray-dried from 90% ethanol were reduced as compared to the unprocessed material (Figure 8A–D). The similarity of the XRD pattern suggested that spray-drying of TAA/ABC resulted in a change in the solid format of TAA in a pattern unrelated to the ABC amount used. The XRD scan confirmed the crystallinity of the product of the spray-drying of TAA with ABC, as is the case when TAA is spray-dried alone. XRD of 1.5% TAA/10% ABC spray-dried from 90% ethanol (Figure 8E) also showed the same XRD peak pattern suggesting that spray-drying of TAA/ABC resulted in a change in the solid state of TAA in a pattern unrelated to the feed concentration used within the specified range (1–1.5% *w*/*v* of TAA).

Particle size was determined using SEM (Table 4). Micronized TAA had an average particle size <5 µm while the mean particle size for the 1% TAA/ABC solutions in 90% ethanol *v*/*v* was 2.26 ± 0.36 µm, 2.45 ± 0.24 µm, and 3.00 ± 0.49 µm for 10, 15, and 20% ABC solutions, respectively. The particle size of the 1.5% TAA from the 10% ABC spray-dried formulation was 2.55 ± 0.65 µm.

The bulk (ρb) and tapped (ρt) densities and Carr’s index of TAA/ABC formulations are summarized in Table 4. The calculated Carr’s index for the spray-dried TAA/ABC formulations indicated the improvement of powder flowability compared to micronized TAA.

### 3.3. Selection of the Appropriate Formula for In-Vivo Studies 

According to the described results, the feasibility of the preparation of an additive-free TAA spray-dried product was confirmed. This conclusion was evidenced by the non-significant differences between spray-dried particles obtained with or without the addition of a blowing agent. Regarding powder flowability and respirability, the same Carr’s index values were obtained for products of corresponding TAA feed concentrations and solvent compositions. Accordingly, the addition of the blowing agent did not contribute to the spray-dried product under the described conditions. This finding was of great potential from the perspective of the pharmaceutical industry as decreasing the number of additives is a key advantage for the preparation for pulmonary delivery to reduce the possible side effects of additive materials on lung tissue. Accordingly, the spray-drying process without the addition of any blowing agent was selected. Minimization of TAA feed concentrations resulted in the minimum particle size, higher porosity, and lower Carr’s index values. Accordingly, the 1% TAA feed concentration was selected.

It was also found that microspheres with the optimal porosity were produced as a result of spray-dried TAA formulations from 90% (*v*/*v*) ethanol without the addition of any blowing agent. Accordingly, the solvent composition of 90% ethanol was selected. 

Solvents played an important role in controlling the morphology of the produced particles. The application of ethanol as a co-solvent and volatilizing agent decreases the solvent vaporization temperature and enables pore formation in the produced particles. It was assumed that during the drying process of the droplets, numerous pores and channels are formed within the structure of the droplet due to the evaporation of the applied solvent. Accordingly, the particles formed from droplets even experience expansion to have higher porosity and lower density.

According to the characterization results, in experiment F1, 1% TAA spray-dried from 90% *v*/*v* ethanol without the addition of a blowing agent was selected for subsequent in vivo studies. 

### 3.4. In Vivo Characterization of TAA Microspheres by the Study of Direct Lung Delivery of Microspheres Particles (M.S.P.s) of TAA Powders by Inhalation

After the formulation of additive-free microspheres containing 100% f TAA by weight, these microspheres possessed physical and aerosol properties suitable for inhalation and transportation to rats through respiratory insufflation as a dry powder. We evaluated the drug concentration in plasma, lung lining fluid (examined by obtaining BALF), and the tissue of lung homogenate.

High-performance liquid chromatography was applied to analyze TAA in the collected in vivo samples. A standard calibration curve of TAA in the mobile phase at λmax of 252 nm showed a linear correlation between the peak area and TAA concentration with a correlation coefficient r = 0.9987 within a concentration range of 0.1–50 µg mL^−1^. The HPLC chromatograms of TAA showed the presence of a major peak at a retention time of approximately 4.0 min in standard and sample chromatograms. Typical HPLC chromatograms of 1 µg/mL TAA in the mobile phase and chromatograms in plasma, BALF, and lung tissues were provided (Figure 9A–D).

The mean plasma concentration (PC) of TAA (µg/mL) versus time in hours is provided in Table 5 and Figure 10A. The mean BALF concentration (FC) of TAA (µg/mL) versus time in hours is provided in Table 5 and Figure 10B. The mean tissue lung concentration (TC) of TAA (µg/mL) versus time in hours after administration of TAA microspheres is shown in Table 5 and Figure 10C. All results are expressed as mean ± S.E.

The pharmacokinetic data presented in Figure 10A–C show that 5 mg of microsphere particles of TAA via insufflation reached the circulation after 15 min from installation to the lungs, the plasma concentration was 0.55 ± 0.02 µg/mL at 15 min, and was 0.03 ± 0.02 µg/mL after 24 h of exposure (Table 5). Calculating TAA inhalation results to the body mass of the rat indicated that the aggregate body dosage is approximately 22 mg/kg through the lungs. Inferable from points of confinement of recognition, we were not ready to quantify the TAA concentration at the BAL liquid past 15 min. At 15 min, we observed this concentration to be (0.67 ± 0.08 µg/mL), and the TAA concentration was (0.06 ± 0.00 µg/mL) at 24 h.

An expected dilution factor was determined to evaluate the relative BAL concentration (i.e., to represent how the BAL acts as a diluent). The collective lung lining liquid volume assessment is 80.2 mL [48]. Since just the right half of the lung was lavaged, it was estimated that 40.1 µL of the lung-covering liquid was diluted at 1 mL from a buffer of BAL. Representing this dilution, the concentration of TAA in the lung-lining liquid after 15 min was determined (16.74 ± 2.0 µg/mL) (Table 5), which was approximately thirty times the calculated concentration of TAA in the plasma, and the concentration of TAA in the fluid of the lung lining at 24 h was estimated to be 1.48 ± 0.07 µg/mL (Table 5). 

The TAA concentration in lung tissue homogenates was determined to be (0.45 ± 0.03 µg/mL) at 15 min and (0.19 ± 0.02 µg/mL) after 24 h. To evaluate the dilution because of the buffer, the left lobe was left, set in 3 mL of buffer, and homogenized (the new volume recorded was 3.16 mL). This way, the lung commitment to the volume is 0.16, and the dilution variable is 0.16/3.16 = 0.0506. The concentration in lung homogenate was found to be (8.96 ± 0.65 µg/mL) at 15 min and (3.79 ± 0.33 µg/mL) at 24 h (Table 5).

These pharmacokinetic data indicate, although less noticeable in the blood or the lung liquid, TAA stays at a higher concentration in lung tissue for no less than 24 h after insufflation. This pharmacokinetic information gives an impression regarding the safety and efficiency of inhaled TAA; however, we believe the outcome to be sufficiently encouraging to strive for further measurement of safety and efficiency testing.

## 4. Conclusions

Additive-free spray-dried TAA microspheres were prepared by a single-step process. Spray-drying of TAA in a concentration range of 1% to 2.5% from a 90% ethanolic solvent resulted in crystalline microsphere particles with a porous surface as indicated by different characterization techniques. Enhanced powder flowability and increased respirability of the product were significantly enhanced via the optimization of feed concentrations and solvent compositions. A lower TAA feed concentration provided greater suitability for pulmonary delivery. The porosity of the formulated microspheres was attained in the described conditions without adding a blowing agent, as the applied solvent played a major role in controlling the morphology of the produced particles. In conclusion, the characterization of the recovered microsphere particles showed favorable characteristics for pulmonary delivery compared to the micronized TAA particles. This warrants future studies for developing spray-dried microsphere formulations of TAA as a therapeutic preparation suitable for pulmonary delivery and opens avenues for applying this technique to formulate pulmonary dosage forms of other medications.

## Figures and Tables

**Figure 1 pharmaceutics-14-02354-f001:**
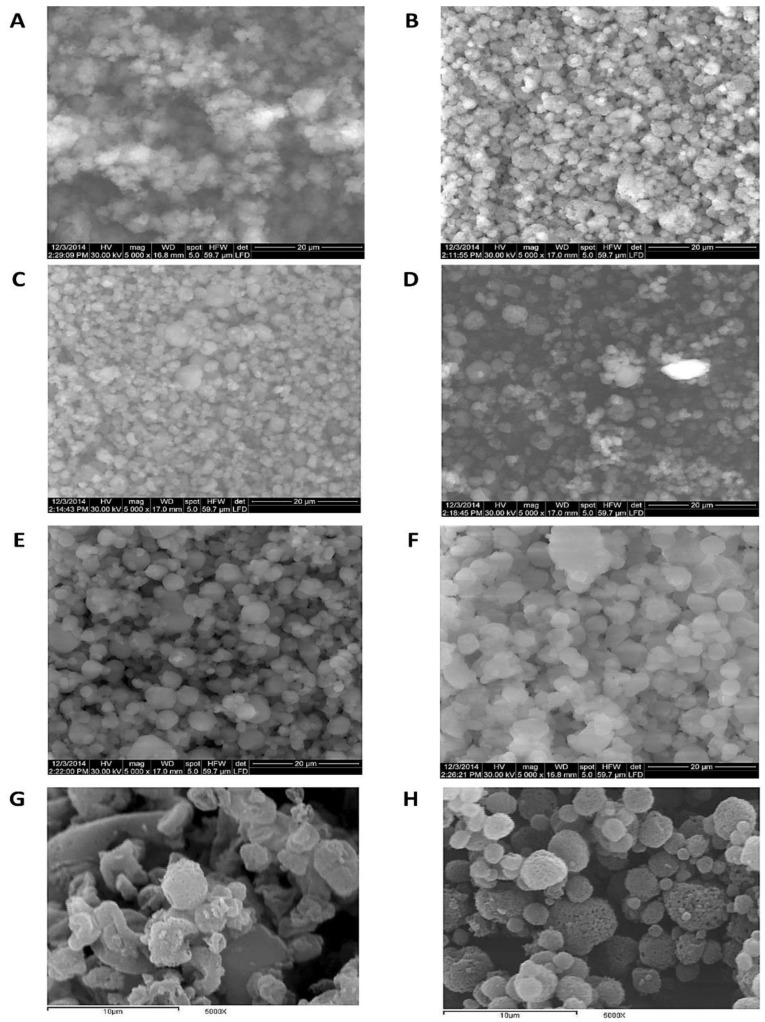
SEM micrographs of different TAA formulations. (**A**) TAA-unprocessed and (**B**) 1%, (**C**) 1.5%, (**D**) 2%, (**E**) 2.5%, and (**F**) 3% (*w*/*v*) TAA systems spray-dried from 90% (*v*/*v*) ethanol, (**G**) 1% TAA system spray-dried from 70% (*v*/*v*) ethanol, and (**H**) 1% TAA system spray-dried from 80% (*v*/*v*) ethanol.

**Figure 2 pharmaceutics-14-02354-f002:**
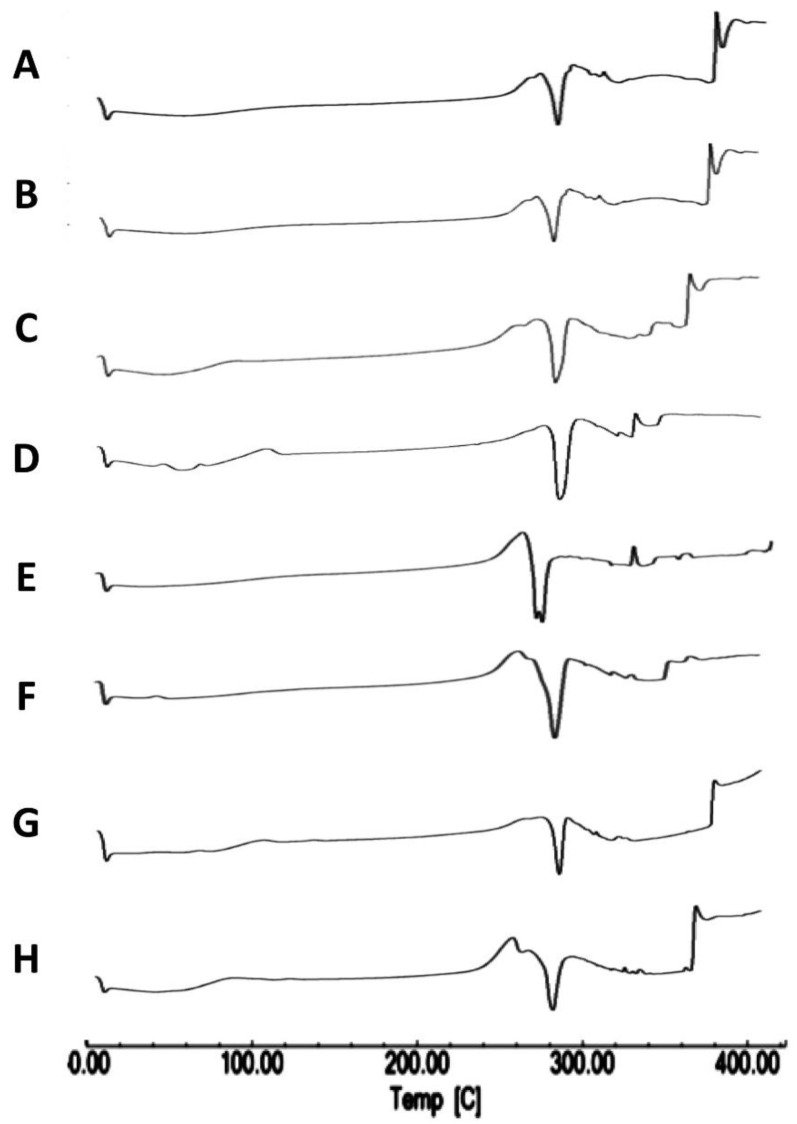
DSC scans of different TAA formulations. (**A**) Micronized TAA raw material unprocessed and (**B**) 1%, (**C**) 1.5%, (**D**) 2%, (**E**) 2.5%, and (**F**) 3% (*w*/*v*) TAA systems spray-dried from 90% (*v*/*v*) ethanol, (**G**) 1% TAA system spray-dried from 70% (*v*/*v*) ethanol, and (**H**) 1% TAA system spray-dried from 80% (*v*/*v*) ethanol.

**Figure 3 pharmaceutics-14-02354-f003:**
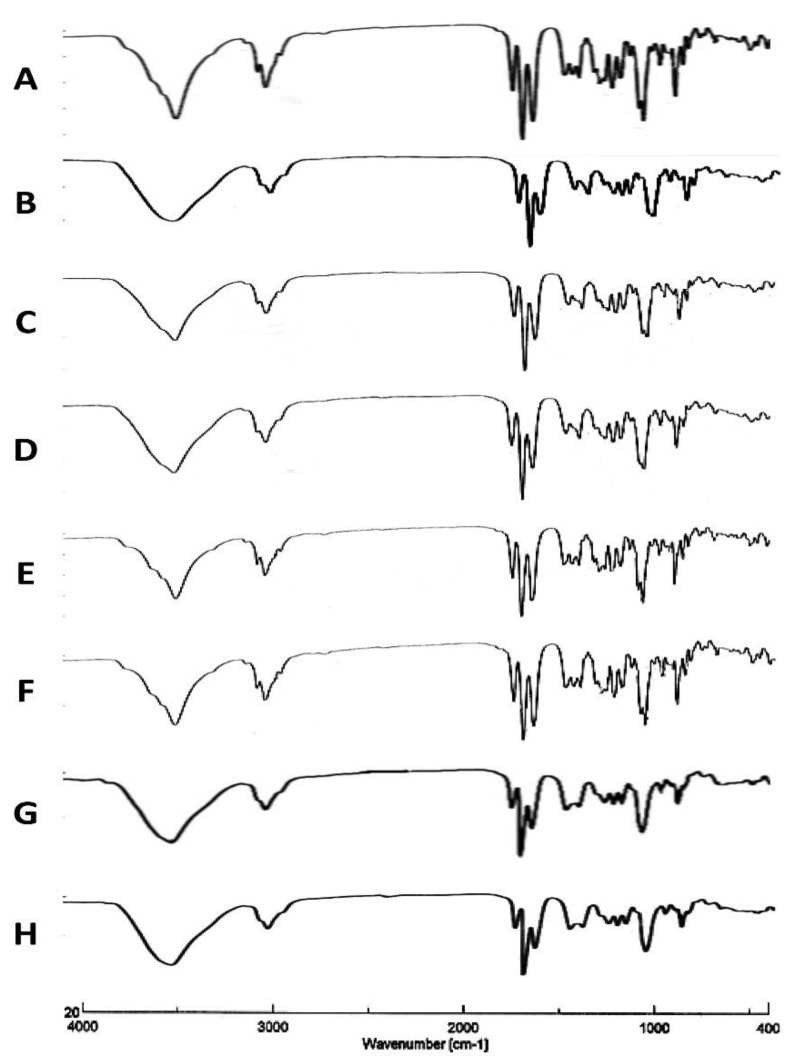
FTIR scans of different TAA formulations. (**A**) Micronized TAA raw material unprocessed and (**B**) 1%, (**C**) 1.5%, (**D**) 2%, (**E**) 2.5%, and (**F**) 3% (*w*/*v*) TAA systems spray-dried from 90% (*v*/*v*) ethanol, (**G**) 1% TAA system spray-dried from 70% (*v*/*v*) ethanol, and (**H**) 1% TAA system spray-dried from 80% (*v*/*v*) ethanol.

**Figure 4 pharmaceutics-14-02354-f004:**
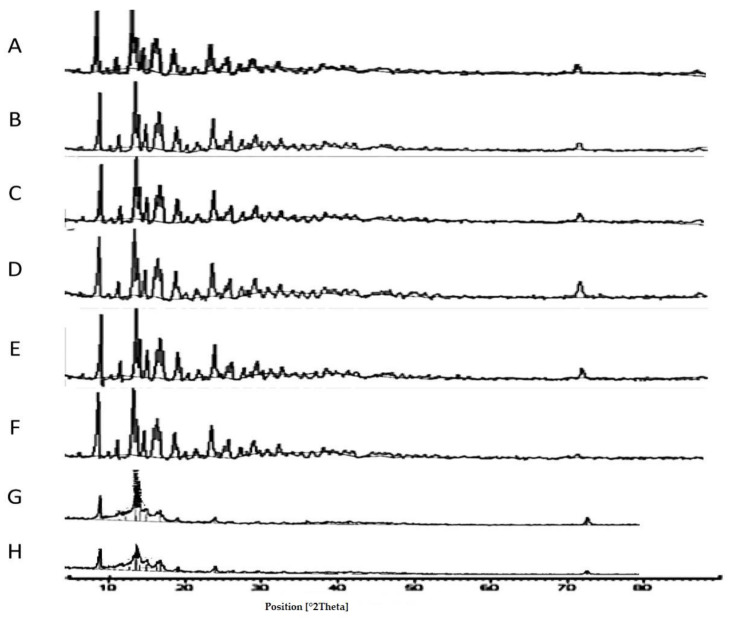
XRD scans of different TAA formulations. (**A**) Micronized TAA raw material unprocessed and (**B**) 1%, (**C**) 1.5%, (**D**) 2%, (**E**) 2.5%, and (**F**) 3% (*w*/*v*) TAA systems spray-dried from 90% (*v*/*v*) ethanol, (**G**) 1% TAA system spray-dried from 70% (*v*/*v*) ethanol, and (**H**) 1% TAA system spray-dried from 80% (*v*/*v*) ethanol.

**Figure 5 pharmaceutics-14-02354-f005:**
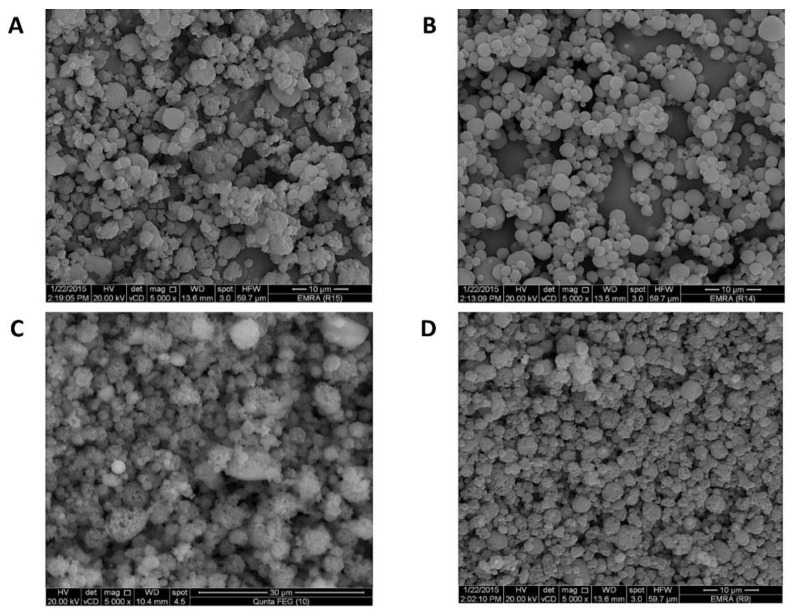
SEM micrographs of different TAA formulations. (**A**) 1% (*w*/*v*) TAA/ammonium bicarbonate (90:10), (**B**) (85:15), and (**C**) (80:20) and (**D**) 1.5 % (*w*/*v*) TAA/ ammonium bicarbonate (90:10) systems spray-dried from 90% (*v*/*v*) ethanol.

**Figure 6 pharmaceutics-14-02354-f006:**
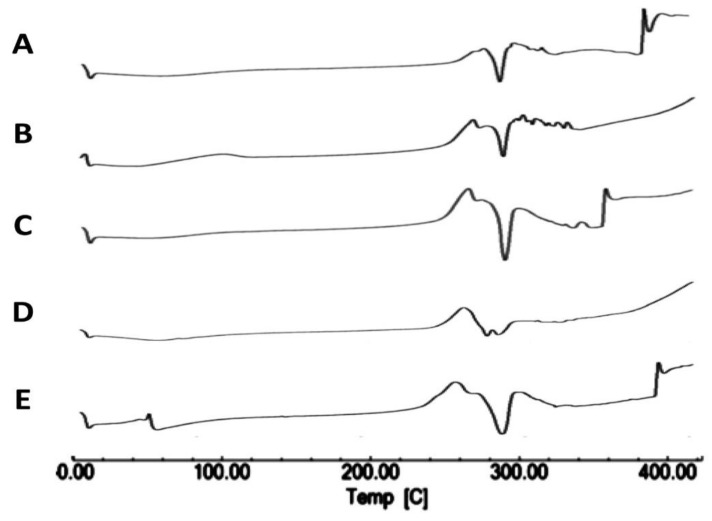
DSC scans of different TAA formulations. (**A**) Micronized TAA raw material unprocessed and (**B**) 1% (*w*/*v*) TAA/ammonium bicarbonate (90:10), (**C**) (85:15) and (**D**) (80:20) and (**E**) 1.5 (*w*/*v*) %. TAA/ammonium bicarbonate (90:10) products spray-dried from 90% (*v*/*v*) ethanol.

**Figure 7 pharmaceutics-14-02354-f007:**
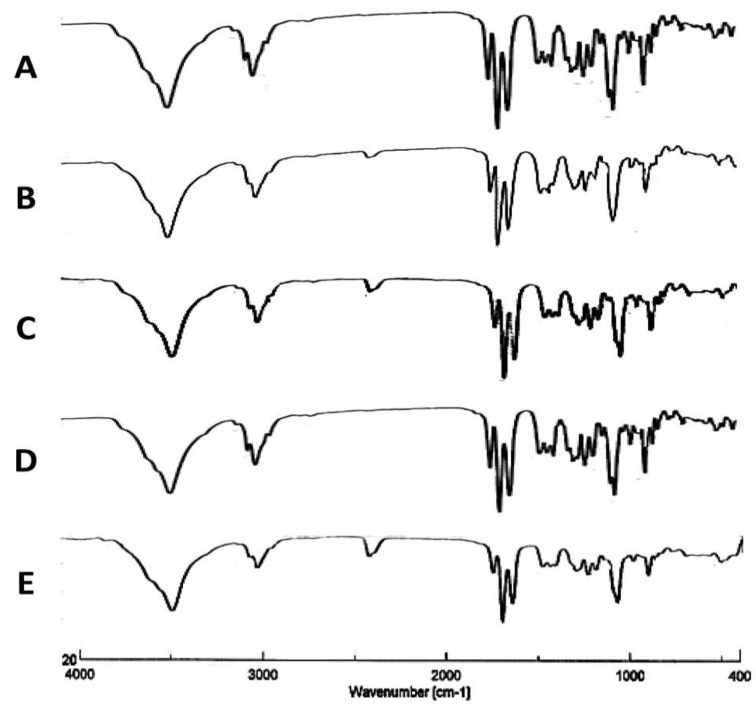
FTIR scans of different TAA formulations. (**A**) Micronized TAA raw material unprocessed and (**B**) 1% (*w*/*v*) TAA/ammonium bicarbonate (90:10), (**C**) (85:15), and (**D**) (80:20) and (**E**) 1.5 (*w*/*v*) % TAA/ ammonium bicarbonate (90:10) products spray-dried from 90% (*v*/*v*) ethanol.

**Figure 8 pharmaceutics-14-02354-f008:**
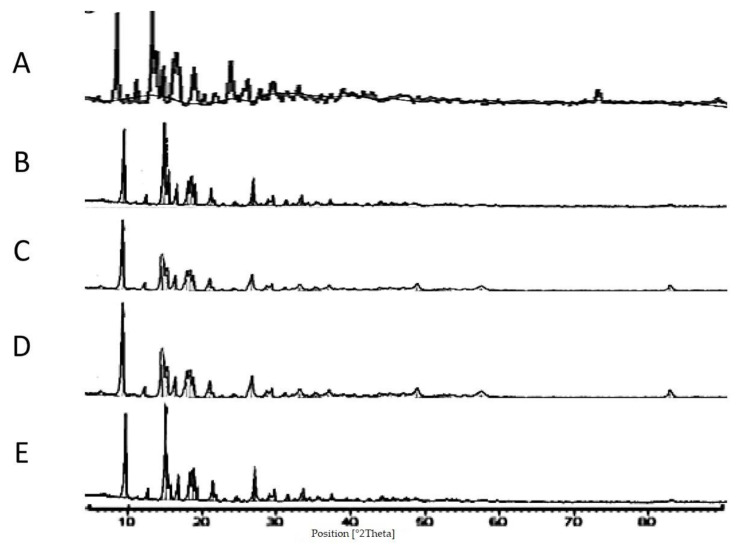
XRD scans of different TAA formulations. (**A**) Micronized TAA raw material unprocessed and (**B**) 1% (*w*/*v*) TAA/ammonium bicarbonate (90:10), (**C**) (85:15), and (**D**) (80:20) and (**E**) 1.5 (*w*/*v*) % TAA/ammonium bicarbonate (90:10) products spray-dried from 90% (*v*/*v*) ethanol.

**Figure 9 pharmaceutics-14-02354-f009:**
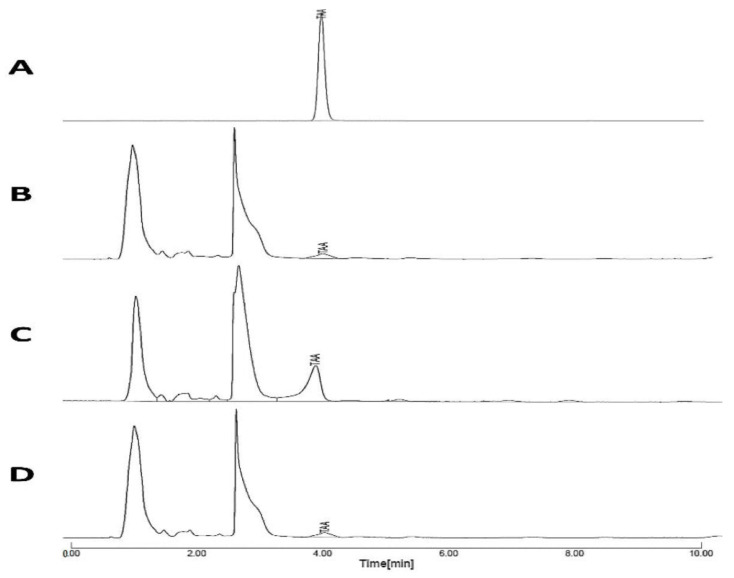
HPLC Chromatograms of (**A**) standard TAA in mobile phase, (**B**) sample of plasma concentration determination, PC, (**C**) sample of lung fluid concentration determination, FC, (**D**) sample of lung tissue determination, TC.

**Figure 10 pharmaceutics-14-02354-f010:**
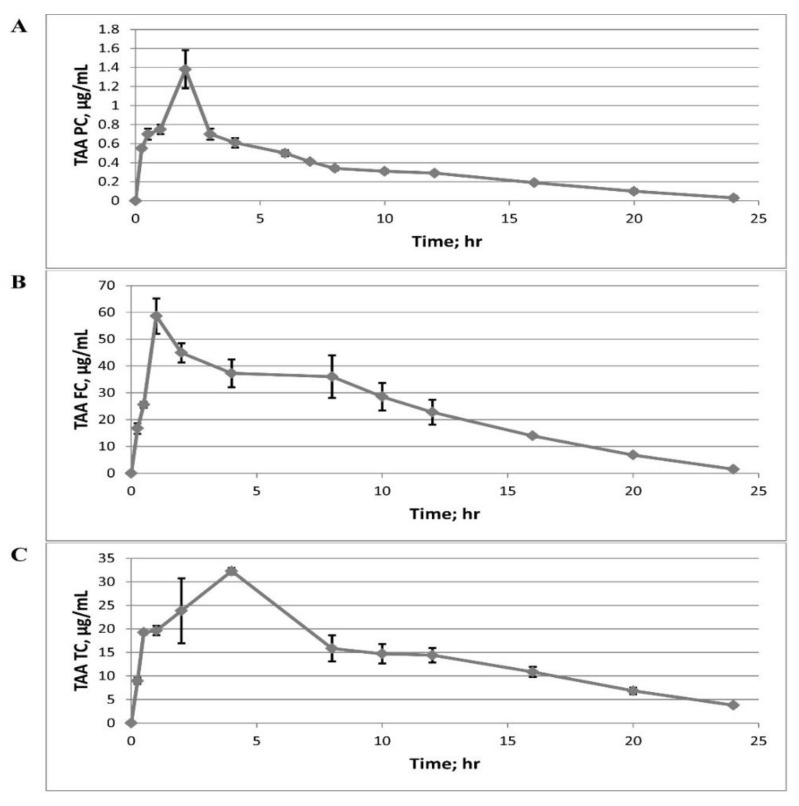
TAA concentration in (**A**) plasma, (**B**) lung fluid, and (**C**) lung tissue after insufflation of 5 mg from dry powders to rats. Mean ± SE, *n* = 3 at each time point.

**Table 1 pharmaceutics-14-02354-t001:** Spray-drying parameters and input parameters for TAA spray-dried products without blowing agent *.

System (Sample ID)	Solvent Composition; Ethanol% *v*/*v* (Volume = 100 mL)	TAA (%*w*/*v*)	Outlet Temp. (°C)	Yield (%)
F1	90%	1%	54–50	39
F2	90%	1.5%	54–48	58
F3	90%	2%	54–50	54
F4	90%	2.5%	52–50	49
F5	90%	3%	50–48	43
F6	70%	1%	50–40	41
F7	80%	1%	52–48	30

* The following parameters were fixed: Inlet temperature at 78 °C, atomizing air flow rate at 670 N L/hr, liquid feed flow rate at 4 mL/min, aspirator at 100% and pump at 40%.

**Table 2 pharmaceutics-14-02354-t002:** Physical examination results of TAA spray-dried products without blowing agent in comparison with micronized TAA.

	D (µm)			ρb			ρt			C		
Formula ID	Mean	±	SE	Mean	±	SE	Mean	±	SE	Mean	±	SE
F1	2.24	±	0.27	0.95	±	0.05	1.18	±	0.07	19	±	0.01
F2	2.52	±	0.44	0.34	±	0.01	0.42	±	0.02	17	±	0.01
F3	3.07	±	0.19	0.39	±	0.01	0.51	±	0.01	24	±	0.00
F4	3.64	±	0.34	0.35	±	0.01	0.42	±	0.02	17	±	0.01
F5	3.93	±	0.44	0.35	±	0.02	0.53	±	0.03	34	±	0.01
F6	2.54	±	0.32	0.42	±	0.02	0.59	±	0.03	29	±	0.01
F7	3.12	±	0.49	1.06	±	0.06	1.34	±	0.09	21	±	0.01
Micronized TAA	99% < 5 um			0.23	±	0.01	0.42	±	0.02	46	±	0.00

D, diameter of sphere representing particle size; ρb, bulk density; ρt, tapped density; C, Carr’s index; SE, standard error.

**Table 3 pharmaceutics-14-02354-t003:** Spray-drying parameters and input parameters for TAA spray-dried products with blowing agent *.

System (Sample ID)	Solvent %*v*/*v* Ethanol, Volume = 100 mL	Solid Conc. (%*w*/*v*)	Blowing Agent; ABC, Conc. (%*w*/*v*)	Outlet Temp. (°C)	Yield (%)
F8	90%	1	10	50–45	30
F9	90%	1	15	53–45	26
F10	90%	1	20	51–48	22
F11	90%	1.5	10	52–45	24

* The following parameters were fixed: Inlet temperature at 78 °C, Atomizing air flow rate at 670 N L/h, Liquid feed flow rate at 4 mL/min, aspirator at 100% and pump at 40%.

**Table 4 pharmaceutics-14-02354-t004:** Physical examination results of TAA spray-dried products with ABC blowing agent in comparison with micronized TAA.

	D (um)			ρb			ρt			C		
Formula ID	Mean	±	SE	Mean	±	SE	Mean	±	SE	Mean	±	SE
F8	2.26	±	0.36	0.96	±	0.04	1.18	±	0.07	19	±	0.01
F9	2.45	±	0.24	1.76	±	0.05	2.59	±	0.08	32	±	0.00
F10	3.00	±	0.49	0.52	±	0.02	0.70	±	0.03	25	±	0.00
F11	2.55	±	0.65	1.32	±	0.07	1.83	±	0.17	28	±	0.03
Micronized TAA	99% < 5 um			0.23	±	0.01	0.42	±	0.02	46	±	0.00

D, diameter of sphere representing particle size; ρb, bulk density; ρt, tapped density; C, Carr’s index; SE, standard error.

**Table 5 pharmaceutics-14-02354-t005:** Measurement and calculation of TAA conc. (µg/mL) in plasma, PC; fluid of lung, FC; and tissue of lung after insufflation, TC. Mean ± SE, *n* = 3.

	PC			FC			TC		
Time (h)	Mean	±	SE	Mean	±	SE	Mean	±	SE
0	0.00	±	0.00	0	±	0	0	±	0
0.25	0.55	±	0.02	16.74	±	2.00	8.96	±	0.65
0.5	0.70	±	0.06	25.56	±	1.04	19.24	±	0.44
1	0.75	±	0.05	58.64	±	6.57	19.64	±	0.98
2	1.38	±	0.20	44.94	±	3.59	23.85	±	6.90
4	0.61	±	0.05	37.26	±	5.21	32.29	±	0.69
8	0.34	±	0.02	36.04	±	7.94	15.85	±	2.77
10	0.31	±	0.02	28.55	±	5.13	14.71	±	2.05
12	0.29	±	0.01	22.78	±	4.61	14.41	±	1.55
16	0.19	±	0.01	13.94	±	0.65	10.86	±	1.07
20	0.10	±	0.02	6.80	±	0.26	6.85	±	0.70
24	0.03	±	0.02	1.48	±	0.27	3.79	±	0.33

## Data Availability

Data are available from M.A. upon request.

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
