# Peer review of "Formulation and Evaluation of Novel Additive-Free Spray-Dried Triamcinolone Acetonide Microspheres for Pulmonary Delivery: A Pharmacokinetic Study"

_pharmaceutics, 2022, doi:10.3390/pharmaceutics14112354_

Round 1

Reviewer 1 Report

The manuscript submited by Shadeed Gad et al. reported a preparation method of TAA microspheres suitable for pulmonary delivery, and chose the optimal TAA formulation #1 to study the pharmacokinetic characters. Overall, the study was interesting, but some parts need to be revised.

1. The introduction part was kinda loose, and cannot provide sufficient background information.  The highlights of preparation method TAA microspheres should be clearly displayed and emphasized.

2. All of the figures cannot be accepted and should be redrawn.

3. There is no discussion in the results part, which should be improved greatly.

Author Response

Authors wish to thank the respective reviewer for his/her effort in revising the manuscript and are pleased to submit their reply.

No

Quotations

Answers

1

The introduction part was kinda loose, and cannot provide sufficient background information.  The highlights of preparation method TAA microspheres should be clearly displayed and emphasized.

The whole introduction was revised for the flow of information and amended by some new parts to highlight the method of TAA preparation as recommended by the respective reviewer.

2

All of the figures cannot be accepted and should be redrawn

All the figures were re-drawn with enhanced resolution in the current revised submission

3

There is no discussion in the results part, which should be improved greatly

Discussion was integrated with results in section 3, Results and discussion

Reviewer 2 Report

In this manuscript, the authors performed Formulation and Evaluation of Novel Additive-Free Spray-Dried Triamcinolone Acetonide Microspheres for Pulmonary Delivery: A Pharmacokinetic Study. In my opinion, some issues should be further addressed and I hope the following comments could be helpful for improving their paper.

1. It would be better if the author add a graphical representation of the overall study as a schematic diagram.

2. kindly revised the abstract and add the main results here.

3. Why different formulations were dissolved in ethanol-water in different ratios. Justify it.

4. The tapped density (ρt) of processed powder was then determined by tapping the cylinder containing the powder 100 times onto a horizontal surface within 1-inch height. Why did 1-inch height keep for this experiment?

5. Kindly add an animal section in the materials and explain it in the detail.

6. Rats were anesthetized by injecting 80 mg/kg of ketamine and xylazine (10 mg/kg). I think these anesthetics are very dangerous and sometimes cause mice death. why did the authors choose this anesthetic agent with an 80 mg/kg dose?

7. SEM images are not clear and they didn't show a scale bar.

8. Figure 3. FTIR scans Figure 4. XRD scans, Figure 5. SEM micrographs, Figure 7. FTIR scans, Figure 8. XRD scans, and Figure 9. HPLC Chromatograms, these figure is not clear, kindly provide neat and clean figures

9.    Discussion: This part requires a thorough development. The authors should clarify the   signalized doubts. They should to demonstrate the advantages and disadvantages of the proposed system against the background of similar systems described earlier. The authors should also present their suggestions related to possible possibilities of practical application of the described solution.

10.      Please revisit the entire manuscript for minor grammar issues. The writing although good needs to be corrected for grammar and sentence construction. I also highly recommend the authors to streamline their writing to keep the underlying conclusions precise and clear . The transitions between ideas seem disconnected. These would only help the reader get more from the review and improve on its quality and appeal

Author Response

Kindly find attached
